# Oocyte Quality Assessment in Breast Cancer: Implications for Fertility Preservation

**DOI:** 10.3390/cancers14225718

**Published:** 2022-11-21

**Authors:** Cristina Fabiani, Antonella Guarino, Caterina Meneghini, Emanuele Licata, Gemma Paciotti, Donatella Miriello, Michele Carlo Schiavi, Vincenzo Spina, Roberta Corno, Mariagrazia Gallo, Rocco Rago

**Affiliations:** 1Physiopathology of Reproduction and Andrology Unit, Sandro Pertini Hospital, 00157 Rome, Italy; 2Department of Obstetrics and Gynecology, Sandro Pertini Hospital, 00157 Rome, Italy; 3Protection of Maternal and Child Health Unit, 02100 Rieti, Italy

**Keywords:** breast cancer, controlled ovarian hyperstimulation, fertility preservation, oocytes quality

## Abstract

**Simple Summary:**

Fertility preservation programs in patients with cancer are one of the most interesting current topics in reproductive medicine. Breast cancer is the most common malignancy in women of reproductive age. Young women with breast cancer are at risk of future infertility, as cancer treatment can be lifesaving, but negatively impacts ovarian function. The clinical risk is related to the age of the patient, the chemotherapy drugs used, and the duration of treatment. Mature oocyte cryopreservation is no longer considered an experimental technique, and many improvements have been made in oocyte cryopreservation. Considering the paucity of research on the effect of breast cancer on the ovarian response in this specific group of cancer patients, we aimed to design a study to investigate the outcome of ovarian stimulation in terms of the oocyte number, maturity, and quality in women with breast cancer.

**Abstract:**

Background: The aim of this study was to evaluate the effects of breast cancer on the ovarian response and on oocyte quality following controlled ovarian hyperstimulation (COH). Methods: This retrospective case-control study evaluated the effects of breast cancer on the ovarian response and on the oocyte quality. Oncological patients with breast cancer undergoing controlled ovarian stimulation cycles for fertility preservation, and age- and date-matched controls undergoing COH for in vitro fertilization (IVF) for male or tubal factor infertility were included in the study. Two hundred and ninety-four women were enrolled: 105 affected by breast cancer and 189 healthy women in the control group. Both groups were comparable in terms of age, BMI, and AMH value. Maximal estradiol levels on the triggering day, duration of stimulation, total amount of gonadotropins administered, number of oocytes retrieved, rate of metaphase 2 oocyte production, and numbers of immature and dysmorphic oocytes were analyzed. Results: Considering factors influencing the oocyte quality, such as age, BMI, AMH, duration of stimulation, E2 level on the triggering day, total FSH cumulative dose, stage, histotype, BRCA status, and hormone receptors, the univariate and multivariate analyses identified breast cancer as a risk factor for the presence of dysmorphic oocytes. Conclusions: The diagnosis of breast cancer does not seem to be associated with the impairment of the ovarian reserve, but is linked to a worsening oocyte quality.

## 1. Introduction

Fertility preservation in female oncology patients should be integrated as part of the management of cancer patients to improve their quality of life. Every day in Italy, about 30 individuals under 40 years of age are diagnosed with cancer, and breast cancer (BC) is the most common malignancy experienced by women undergoing fertility preservation treatment [1]. The survival rates of these patients, who are often of reproductive age, have been steadily increasing due to the greater efficacy of novel oncological therapies. These data, associated with the trend of delayed first pregnancy in these women, requires an accurate evaluation and prevention of gonadotoxic damage through counseling on the options for fertility preservation and future reproductive planning [2]. In fact, the deleterious impact of oncological medical therapy and surgical approaches on fertility are well known. The risk of loss of ovarian function by chemotherapy is related to the age of the patient, the chemotherapy drug used, and the duration of treatment. Furthermore, oocytes are very susceptible to abdominal or pelvic radiotherapy [2].

In 2013, the American Society of Clinical Oncology Guidelines recommended mature oocyte cryopreservation as a standard technique for fertility preservation in young women, providing a practical option for women who do not have a male partner and for teenagers [3]. Ovarian tissue cryopreservation for future transplantation is the best program for prepubertal girls, and currently, in vitro maturation of oocytes is an experimental procedure [3]. Oocyte vitrification has become a routine human-assisted reproductive technology (ART) technique. Studies on reproductive cells from different species indicate that cryopreservation can induce oocyte alterations, such as alterations of mitotic apparatus or sublethal injuries, such as oxidative stress-induced DNA damage, altered metabolism, and transcription and translation abnormalities that may not be detectable morphologically. Mammalian oocytes a have large cytoplasm with abundant mitochondria in the ooplasm that can experience structural and functional damage during the cryopreservation process. Therefore, a reduction in viability and post-thaw developmental competence can be correlated with the extent of mitochondrial damage in oocytes [4,5]. IVF using vitrified oocytes could produce similar fertilization and pregnancy rates to IVF with fresh oocytes [3]. 

However, studies on the outcome of the ovarian response after ovarian stimulation in specific cancer patients are limited. Some reports have suggested a deleterious impact of the oncological disease with a lower response to ovarian stimulation [6,7,8,9,10,11,12]. Moreover, few reports have suggested a deleterious impact of the oncological disease on the quality of follicular growth and ovarian function [13,14,15]. The aim of our study was to analyze the outcomes of ovarian stimulation in women undergoing fertility preservation by investigating the effect of breast cancer on the oocyte quality, mainly in terms of the metaphase II (MII) total oocyte ratio and the dysmorphic oocyte ratio. 

## 2. Materials and Methods

This was a retrospective, single-center, case-control study carried out in the Department of Reproductive, Pathophysiology, and Andrology at the Regional Reference Center on Fertility, Oocyte Bank of the Sandro Pertini Hospital, Rome, Italy. It was approved by the local Ethical Committee (IRB protocol number: 0104898/2020, date of approval: 16 June 2020). Informed consent regarding the research use of medical information was obtained from all individual participants included in the study in accordance with local and international legislation (Declaration of Helsinki) [16]. 

### 2.1. Study Population 

From June 2016 to May 2021, patients undergoing fertility preservation at our center were enrolled. The case group included breast cancer patients, and the control group included healthy patients. In order to avoid bias, controls were selected according to age- and pick-up date-matching criteria. Breast cancer patients were staged according to the American Joint Committee on Cancer (AJCC–III edition) staging system [17]. The inclusion criteria were as follows: women of reproductive age (aged 18–38 years); individuals with tubal or male infertility (for the control group); oncology patients affected by breast cancer who had not yet started chemotherapy or radiotherapy treatment (for the case group); and a signed written informed consent and signature prior to the health data treatment. The exclusion criteria were as follows: aged over 38 years; women with benign ovarian pathologies that compromise fertility; Premature Ovarian Failure (POF) risk with an AMH value of <0.7 ng/mL; endometriosis; infertility due to ovarian diseases; previous ICSI cycles; previous risk factors for thromboembolic events; previous ovarian surgery; and chemotherapy or radiotherapy started before ovarian stimulation. The data recorded included demographics, type of cancer, gonadotropin-starting dose, maximal estradiol (E2) levels, duration of stimulation, total amount of gonadotropins administered, and the number and quality of oocytes retrieved. For all patients, anthropometric, clinical, and biological parameters were analyzed. The anthropometric parameters collected were the age and body mass index (BMI). The clinical characteristics analyzed were the AMH levels measured using Elecsys^®^ AMH (Anti-Mullerian Hormone)-ROCHE and routine blood chemistry, and an anesthesiologic examination was conducted. Finally, an antral follicle count was carried out for every patient, but this was not used as a parameter to determine the ovarian reserve in this study, because it was performed by different operators. 

### 2.2. Ovarian Stimulation Protocol

In women considered suitable, ovarian-controlled stimulation with an antagonist protocol was performed. A recombinant gonadotropin FSH (follicle-stimulating hormone) with a maximum dosage of 300 IU was used, starting from the second bleeding day of a physiological menstrual cycle or according to a “Random start” protocol if the dates did not coincide with the period available, in cancer patients for the start of medical and/or surgical therapy. The gonadotropin dose was determined based on the patient’s age and AMH value. In patients with sensitive estrogen tumors, Letrozole was administered from the second day of stimulation with a dosage of 5 mg per day. Subsequently, hormonal dosages E2 (17-beta-estradiol) and PR (progesterone) and transvaginal ultrasound were performed every 2/3 days to evaluate the ovarian response and avoid the possibility of ovarian hyperstimulation. Pick-up for oocyte recovery was performed 36 hours after the induction of ovulation. In the control group, ovulation was induced by the administration of recombinant alpha choriogonadotropin. In cancer patients, the analog of GnRH (gonadotropin-releasing hormone) was used at a dose of 0.2 mL. After oocyte collection, the oocytes were stripped of the oophorus cumulus and the radiated crown. 

The biological parameters examined after oocyte recovery included the total number of oocytes, the number of mature metaphase II (MII) oocytes, the number of immature oocytes (metaphase I oocytes, germinal vesicles oocytes), and the number of dysmorphic oocytes. Oocyte abnormalities were divided into extracytoplasmic and intracytoplasmic. Extracytoplasmic abnormalities included shape abnormalities (irregular shape of the MII oocyte), zona pellucida abnormalities, and perivitelline space abnormalities (large PVS and PVS granularity). Intracytoplasmic abnormalities included different types and degrees of cytoplasmic granulations and variations in the color and appearance of refractile bodies, smooth endoplasmic reticulum clusters, or vacuolization in the ooplasm. The mature and suitable oocytes underwent vitrification and were stored at –196 °C in the Cryobank in special containers containing liquid nitrogen.

### 2.3. Statistical Analysis

Case and control groups were compared in terms of baseline and cycle parameters (age, AMH, starting dose of gonadotropins, total amount of gonadotropins administered, and number of stimulation days) and also by ovarian responses (maximal E2 and PR levels on trigger day and number of oocytes retrieved). Subsequently, the two groups were compared in terms of oocyte biological parameters (total number of oocytes, number of mature metaphase II (MII) oocytes, number of immature oocytes, number of dysmorphic oocytes). Comparisons were made with the two-sided Pearson’s chi square and Mann–Whitney U tests, as appropriate. Categorical variables are reported as *n* (%), while continuous variables are described using the mean ± standard deviation (SD) and median (first quartile, Q1–third quartile, Q3). The normality of continuous distributions was assessed with the Shapiro–Wilk test. The survival outcomes for the BC patients were evaluated according to the recurrence occurrence and the follow-up interval was calculated as the time elapsed from the pick-up date and the date of last follow-up visit. Uni- and multivariable logistic regressions were applied in order to assess the impact of breast cancer on the presence of dysmorphic oocytes. Age, BMI, AMH, duration of stimulation, E2 level at triggering day, and the total FSH cumulative dose were considered factors influencing the oocyte quality, as suggested by the literature. In order to assess the impact of the abovementioned variables on the presence of dysmorphic oocytes, a subgroup analysis was also performed for the case and control patients. The stage, histotype, BRCA status, and hormone receptors were also taken in account for the BC patients. The number of dysmorphic oocytes and the percentage of dysmorphic oocytes with respect to the total number of retrieved oocytes per patient were also evaluated with uni- and multivariable linear regression analyses. A statistical analysis was performed using STATA software (STATA/BE 17.0 for Windows, StataCorp LP, College Station, TX 77845, USA). Two-sided tests were applied, and the significance level was set at *p* < 0.05. No imputation was carried out for missing data.

## 3. Results

Two hundred and ninety-four women were enrolled in the study: 105 women affected by breast cancer in the case group and 189 healthy women in the control group. Both groups were comparable in terms of age, BMI, and AMH value (Table 1). The pathological and clinical characteristics of the breast cancer patients are summarized in Table 2. 

Fifty-one patients had a stage I breast cancer diagnosis according to the American Joint Committee on Cancer (AJCC) 2017 classification and were subsequently classified as low-stage disease patients; 49 patients had a stage II–III breast cancer diagnosis and were classified as high-stage disease patients. Triple-negative patients accounted for 14.3% of the population. Data on cancer grade were available for 95 (90.5%) patients. There were 51 patients suffering from G3 breast cancer (48.6%). The median age was 33 years in the cancer group (range, 32–36 years) and 34 years in control group (range, 32–37 years) (*p* = 0.059). There were no significant differences in terms of basal fertility indices between the two groups: the median AMH levels were 2.3 ng/mL in the BC patients and 2.8 ng/mL in the control group (*p* = 0.103). The other baseline demographic and clinical characteristics of the patients are shown in Table 2. The median length of stimulation was 11 days in both groups. The median of total dose of gonadotropins (FSH cumulative dose) used was significantly higher in the cancer patients than in the healthy women (2250 vs. 1425, *p* < 0.0001), and on the triggering day, the median E2 level was significantly lower in the study group vs. the control group (317.5 vs. 1043.5, *p* < 0.0001). A total of 15.5% of patients did not receive aromatase inhibitors, because they did not express estrogen and progesterone receptors. The median number of total retrieved oocytes was 13 in the group of women with cancer and 7 in healthy women (*p* < 0.0001). The median number of total mature oocytes (in metaphase II) was eight in the study group vs. six in the control group (*p* < 0.0001) (Table 3). The median number of total immature oocytes (oocytes MI + germinal vesicle) was two in cancer patients and none in the control group (*p* < 0.0001). The median number of total dysmorphic oocytes was one in the cancer group vs. none in the control group (*p* < 0.0001) (Table 3). 

Figure 1 shows the main anomalies recognized in the recovered dysmorphic oocytes and Figure 2 shows the comparison between a denuded MII oocyte and a dysmorphic oocyte. The multivariable analysis identified cancer as a risk factor for the presence of dysmorphic oocytes (OR (95%CI):3.92 (1.84–8.35)) (Table 4). 

Moreover, in both the case and control groups, age, BMI, AMH, duration of stimulation, E2 level on the triggering day, and total FSH cumulative dose, and for BC patients only, stage, histotype, BRCA status, and hormone receptors were not statistically significantly associated with the presence of dysmorphic oocytes (Table 5 and Appendix A). 

Finally, our data confirm that the cancer is the only risk factor, not only for the presence of dysmorphic oocytes, but also with respect to the number of dysmorphic oocytes and to the percentage of dysmorphic oocytes with respect to the total number of retrieved oocytes for each patient (Appendix A). 

At the time of writing this paper, two patients had achieved spontaneous pregnancies and were waiting to give birth. At 24 months from FP, only two patients (stage I, one G2 and the other G3, Luminal A, BRCA wild-type) had experienced a recurrence of breast cancer (1.9%) at a median of 27 months after FP. The median follow-up period for BC patients was 26.8 months (min–max: 3.9–58.4) All patients showed no evidence of disease at the last follow-up visit.

## 4. Discussion

Our study confirmed that the cryopreservation of mature oocytes is safe and effective given the number of mature retrieved oocytes, and it is now the standard fertility-preserving procedure for BC patients, regardless of receptor status [18]. Moreover, our study showed that cancer is associated with a four-fold increase in risk for the presence of dysmorphic oocytes compared with the control group as well as for the retrieval of immature oocytes. On the contrary, the stage of BC does not negatively affect the quality of retrieved oocytes. BC represents the most frequent oncological diagnosis in women during the reproductive years. Unfortunately, BC survivors have a low chance of pregnancy after diagnosis compared with their normal counterparts, especially when adjuvant therapy is prescribed [19,20]. 

In the literature, cryopreservation of oocytes after controlled ovarian stimulation is described as the standard strategy for fertility preservation in women with BC [21]. A multidisciplinary approach, including the participation of several clinical experts, such as an oncologist, gynecologist, and psychologist, is essential during the implementation of treatment to improve the standard of care and achieve desired family planning in oncological patients. In particular, the assessment of the psychological status is very important for measuring the levels of post-traumatic stress symptoms, depression, and anxiety in these women [22,23].

As demonstrated in our study, several studies have reported that ovarian stimulation is safe with regard to lack of relapse and overall survival in this setting [24]. The present study showed that the ovarian reserve is not adversely affected in patients with BC. In fact, serum AMH levels were overlapping between the case and control groups with no statistically significant difference. According to our data, most recent studies did not demonstrate significant differences in the ovarian reserve between the cancer and control groups [6,8]. Other authors, in the past, showed a lower response in female oncology patients and a significant relationship between the age of the woman and the total number of oocytes retrieved. 

Garcia-Velasco et al. failed to evaluate the baseline ovarian reserve in women with cancer and controls, but this feature is necessary to distinguish the ovarian reserve and the response to stimulation between these groups [12]. A previous meta-analysis of seven studies with a total of 218 cancer patients suggested that there is a poorer response in cancer patients compared with age-matched controls. However, these studies had several limitations, because they included different ovarian stimulation protocols and different inclusion criteria [11]. Of interest is the finding of lower E2 levels in our study group versus the control group, as described previously by Almog et al. [9], despite the higher doses of recombinant FSH used in cancer patients, although this result is partly related to the use of aromatase inhibitors during ovarian stimulation in women with breast cancer. Some authors have demonstrated that aromatase inhibitors may modify the follicular fluid by promoting early development of the antral cavity of follicles [25,26]. However, other authors concluded that the cancer state has a possible deleterious effect on the granulosa cells, the main source of E2, resulting in reduced levels of E2. Indeed, some studies have found that low estradiol levels, regardless of the use of aromatase inhibitors, could be related to a maturation defect of oocytes and to impaired oocyte quality. Additionally, the increased catabolic state and increased stress hormone levels associated with neoplastic disease may adversely affect the ovarian reserve and oocyte quality [13,14]. In the literature, few reports have focused on the pattern of oocyte quality during ovarian stimulation in female oncology patients [6,13].

In agreement with our results, Decanter et al. observed that the number of immature oocytes, particularly dysmorphic oocytes, was significantly higher in cancer patients [13]. These findings are consistent with those of other authors who hypothesized the presence of a possible relationship between a lower oocyte quality and inflammatory, endocrinological, and immune modifications; a modified cytokine network; apoptotic processes; catabolic state; and increased stress hormone levels in neoplastic patients [14,27,28]. Regarding the defect of oocyte maturation shown in our study, according to Decanter et al., the possibility of unfavorable consequences of ovulation induction by the gonadotrophin-releasing hormone agonist (GnRHa) occurring in female oncology patients cannot be excluded. Some studies on GnRHa triggering during conventional IVF treatment for fertility preservation in women with breast cancer or for oocyte donation programs have found at the least the same or better maturation rates for retrieved oocytes [13]. Only a few studies have separately analyzed the influences of the various types of cancer on the ovarian stimulation response and on oocyte quality [7,8,9,13].

Overall, in this study, we found a good response to ovarian stimulation in breast cancer patients, but there were significantly higher numbers of dysmorphic oocytes and immature oocytes than in the control group undergoing IVF for male factor infertility. In fact, our univariate and the multivariate analysis identified that cancer is the only risk factor for the presence of dysmorphic oocytes in BC patients. However, these women did not have a compromised ovarian reserve in agreement with the results of several authors [6,9,14,29,30]. In contrast to our results, Quinn et al. showed that breast cancer diagnosis is not associated with a decreased maturity rate of retrieved oocytes compared with patients undergoing elective fertility preservation for male factor [8]. All patients underwent the same ovarian stimulation protocol. Our study was limited because all cancer patients received the GnRH agonist for the induction of final maturation, while the controls received recombinant alpha choriogonadotropin. In BC patients, we used a GnRHa trigger because it can be effective for the induction of oocyte maturation and prevents ovarian hyperstimulation syndrome (OHSS) during fertility preservation cycles using an antagonist protocol [1,31]. Recent studies have suggested that donors who receive a GnRH agonist trigger versus a human chorionic gonadotropin (hCG) trigger have a similar number of retrieved oocytes, percentage of metaphase II oocytes, and rates of fertilization, implantation, and pregnancy, but a significantly decreased rate of OHSS [32]. In addition, our data show, despite the small sample size, that the stage of breast cancer does not influence the number of retrieved dysmorphic oocytes. In agreement with our results, Cioffi et al. demonstrated that the stage and grade of breast cancer do not impact the number of retrieved mature oocytes [33]. Furthermore, the strength of our study is the importance and specificity of the topic. The limitations of our study concern the paucity of specific cancer groups, especially BRCA mutated breast cancer patients, although oocyte cryopreservation is a safe and valid strategy for fertility preservation in BRCA carriers and involves the utilization of different triggers for the induction of final maturation. We still do not have follow-up data to evaluate the competence of vitrified MII oocytes for oncology patients, and we cannot report information on spontaneous births. Moreover, we still have a few patients in our population with BRCA1 and BRCA2 mutations who are being studied to determine ovarian reserve and fertility preservation outcomes. Limited data are available on the possible deleterious impact of BRCA mutations on the outcome of ovarian stimulation for fertility preservation in women with breast cancer [34,35]. 

## 5. Conclusions

In conclusion, the diagnosis of breast cancer does not seem to be associated with an impairment of the ovarian reserve, but with a worsening of oocyte quality. However, our multivariate analysis identified cancer diagnosis as being associated with a four times greater risk of retrieving dysmorphic oocytes. Certainly, proposing a cryopreservation path to these patients is important to allow them a chance of becoming pregnant in the future. Further studies are necessary to evaluate the long-term outcomes, clinical implications, fertilization rates, pregnancy rates, and the etiopathogenetic mechanisms underlying oocyte abnormalities in this specific group of female oncology patients. 

## Figures and Tables

**Figure 1 cancers-14-05718-f001:**
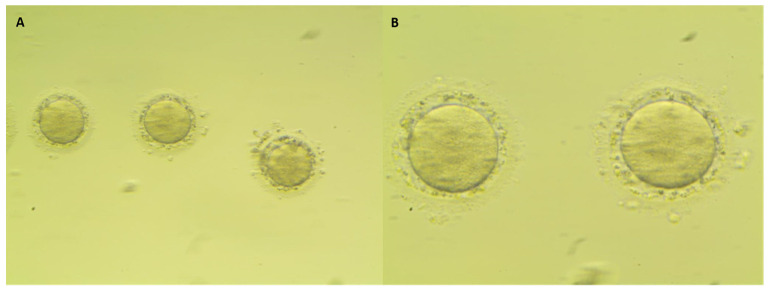
Details of dysmorphic oocytes: (**A**) cytoplasmic dimorphism with a burned-look alteration and (**B**) large perivitelline space with granules. Oocytes showed fragmented cytoplasm ((**A**): 200×; (**B**) 400×).

**Figure 2 cancers-14-05718-f002:**
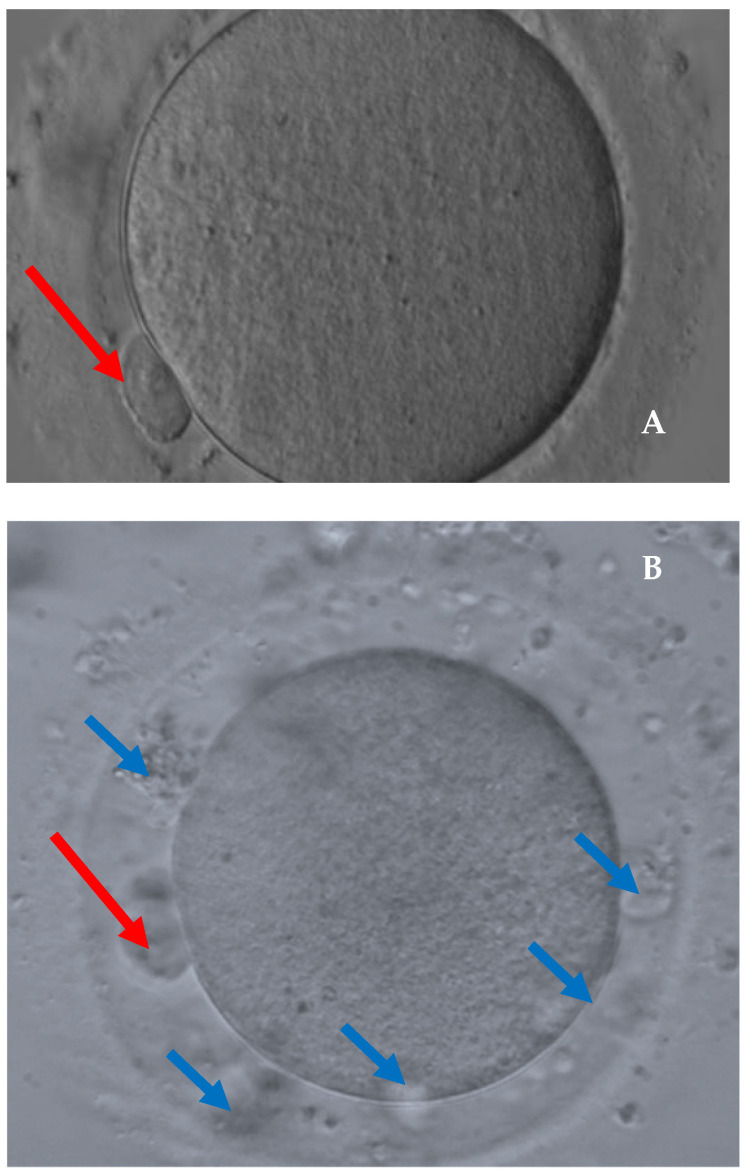
Comparison between a denuded MII oocyte (**A**) and a dysmorphic oocyte (**B**). 400× magnification. (**A**): Intact polar body in the perivitelline space (red arrow) and normal homogeneous cytoplasm. (**B**): Fragmented polar body (red arrow) and enlarged perivitelline space with granules (blue arrows); granular cytoplasm.

**Table 1 cancers-14-05718-t001:** Demographic and clinical characteristics of the study population according to the type of patient (cancer versus control).

Characteristic	All Cases *n* = 294	Cancer Patients *n* = 105	Control Group *n* = 189	*p*-Value
Age, years				
Median (Q1–Q3)	34 (32–36)	33 (32–36)	34 (32–37)	0.059
Mean ± SD	33.6 ± 3.4	33.1 ± 3.6	33.9 ± 3.3	
BMI, kg/m^2^				
Median (Q1–Q3)	21.8 (20.1–24.5)	21.3 (19.8–23.9)	21.9 (20.2–25)	0.119
Mean ± SD	22.6 ± 3.5	22.2 ± 3.4	22.9 ± 3.6	
AMH, ng/mL				
Median (Q1–Q3)	2.6 (1.6–4.1)	2.3 (1.3–3.7)	2.8 (1.8–4.3)	0.103
Mean ± SD	3.2 ± 2.4	3 ± 2.4	3.3 ± 2.3	
E2 peak at triggering, pg/mL				
Median (Q1–Q3)	705 (370.5–1350.3)	317.5 (198.3–690)	1043.5 (597.8–1495.8)	**<0.0001**
Mean ± SD	920.3 ± 697.6	522.1 ± 572.1	1140.5 ± 663.4	
PR peak at triggering, ng/mL				
Median (Q1–Q3)	1.2 (0.7–1.9)	1.6 (1.1–2.5)	0.9 (0.6–1.6)	**<0.0001**
Mean ± SD	1.8 ± 2.7	2.9 ± 4.2	1.2 ± 0.7	
FSH cumulative dose, IU				
Median (Q1–Q3)	1725 (1200–2400)	2250 (1700–3000)	1425 (1050–1950)	**<0.0001**
Mean ± SD	1832.9 ± 944.2	2377.3 ± 994.3	1527.1 ± 761.9	
Duration of ovarian stimulation, days				
Median (Q1–Q3)	11 (10–12)	11 (10–12)	11 (10–12)	0.475
Mean ± SD	11.1 ± 1.9	11.1 ± 2.5	11.1 ± 1.6	

Results are presented as the median (Q1–Q3) and mean ± SD. *p*-values were calculated with the two-sided Mann–Whitney U test for continuous variables (not normally distributed). Bold font highlights statistically significant differences. Q1: first quartile. Q3: third quartile. SD: standard deviation. BMI: body mass index. AMH: anti-Mullerian hormone. E2: estradiol. PR: progesterone. FSH: follicle-stimulating hormone.

**Table 2 cancers-14-05718-t002:** Pathological and clinical characteristics of the 105 cancer patients included in the study.

Characteristic	Breast Cancer Patients *n* = 105
Stage	
I	51/100 (51.0)
II	32/100 (32.0)
III	17/100 (17.0)
Histotype	
Duttal	94/103 (91.3)
Lobular	4/103 (3.9)
Other	5/103 (4.9)
BRCA	
Wild-type	74/95 (77.9)
Mutated	19/95 (20.0)
VUS *	2/95 (2.1)
Grade of differentiation	
1	5/95 (5.3)
2	39/95 (41.1)
3	51/95 (53.7)
BRCA mutation type	
BRCA1	7/21 (33.3)
BRCA2	6/21 (28.6)
Unknown	8/21 (38.1)
Hormone receptors	
ER+ and PR+	75/103 (72.8)
ER− and PR+	0/103 (0)
ER+ and PR−	10/103 (9.7)
ER− and PR−	18/103 (17.5)
HER-2 expression	
Absent	83/100 (83.0)
Present	17/100 (17.0)
Triple negative	13/99 (13.1)
Recurrence	2 (1.9)
Pregnancy	
No	102/104 (98.1)
Spontaneous	2/104 (1.9)
Assisted fecundation	0/104 (0)

Results are presented as *n* (%). BRCA, breast cancer gene. VUS, variants of uncertain significance. ER: estrogen receptor. PR: progesterone receptor. HER-2: human epidermal growth factor receptor 2. * BRCA2.

**Table 3 cancers-14-05718-t003:** Quality of oocytes in the study population according to the type of patient (breast cancer versus control).

Characteristics	All Cases *n* = 294	Breast Cancer Patients *n* = 105	Control Group *n* = 189	*p*-Value
Retrieved oocytes (mean ± SD)				
Median (Q1–Q3)	8 (5–13)	13 (8–17)	7 (5–10)	**<0.0001**
Mean ± SD	9.9 ± 6.4	13.3 ± 7.3	7.9 ± 4.8	
MII mature oocytes				
Presence	290 (98.6)	103 (98.1)	187 (98.9)	0.548
Median (Q1–Q3)	6 (4–10)	8 (5–13)	6 (3–7)	**<0.0001**
Mean ± SD	7.2 ± 4.7	8.9 ± 5.2	6.3 ± 4.1	
Immature oocytes (MI + germinal vesicle)				
Presence	167 (56.8)	76 (72.4)	91 (48.1)	**<0.0001**
Median (Q1–Q3)	1 (0–2)	2 (0–4)	0 (0–2)	**<0.0001**
Mean ± SD	1.7 ± 2.6	2.9 ± 3.6	1.1 ± 1.6	
Dysmorphic oocytes				
Presence	119 (40.5)	58 (55.2)	61 (32.3)	**<0.0001**
<25%	86/119 (72.3)	40/58 (69.0)	46/61 (75.4)	**0.433**
≥25%	33/119 (27.7)	18/58 (31.0)	15/61 (24.6)	
Median (Q1–Q3)	0 (0–1)	1 (0–2)	0 (0–1)	**<0.0001**
Mean ± SD	0.9 ± 1.8	1.5 ± 2.3	0.6 ± 1.3	
Percentage *				
Median (Q1–Q3)	0 (0–14.3)	5.3 (0–20.8)	0 (0–11.1)	**0.0003**
Mean ± SD	8.7 ± 14.8	11.5 ± 14.7	7.2 ± 14.7	

Results are presented as *n* (%) except where indicated. *p*-values were calculated with Pearson’s chi square test for categorical variables and with the Mann–Whitney U test for continuous variables (not normally distributed). Bold font highlights statistically significant differences. SD: standard deviation. Q1: first quartile. Q3: third quartile. M: metaphase. * Number of dysmorphic oocytes/total number of retrieved oocytes * 100.

**Table 4 cancers-14-05718-t004:** Impacts of variables on the oocyte quality: uni- and multivariable logistic regression according to the presence of dysmorphic oocytes.

Characteristic	Patients at Risk	N° of Events	Univariable Analysis		Multivariable Analysis *	
			Odds Ratio (95% CI)	*p*-Value	Odds Ratio (95% CI)	*p*-Value
Breast cancer	294	119	2.59 (1.58–4.23)	< 0.0001	3.92 (1.84–8.35)	< 0.0001
Age						
Continuous	294	119	0.94 (0.88–1.01)	0.094	0.97 (0.89–1.05)	0.425
≤34 years	162	76	1 (Ref)		NI	-
>34 years	132	43	0.55 (0.34–0.88)	0.013		
BMI	270	114	1.00 (0.93–1.07)	0.990	1.00 (0.92–1.08)	0.995
AMH	248	100	1.06 (0.95–1.18)	0.264	1.05 (0.93–1.19)	0.409
Duration of stimulation	292	117	1.03 (0.91–1.16)	0.632	1.05 (0.90–1.24)	0.514
E2 level at triggering day	292	118	1.00 (0.99–1.01)	0.553	1.00 (1.00–1.01)	0.033
Total FSH cumulative dose	292	118	1.00 (0.99–1.01)	0.254	1.00 (0.99–1.00)	0.641

Bold font highlights statistically significant difference. CI: confidence interval. NI: not included. BMI: body mass index. AMH: anti-Mullerian hormone. E2: estradiol. FSH: follicle-stimulating hormone. * 225 patients at risk and 92 events. Age was included in the multivariable analysis as a continuous variable.

**Table 5 cancers-14-05718-t005:** Impacts of variables on oocyte quality according to the presence of dysmorphic oocytes in the group of breast cancer patients.

Characteristic	Univariable Logistic Regression			
	Patients at Risk	N° of Events	Odds Ratio (95% CI)	*p*-Value
Age				
Continuous	105	58	0.95 (0.85–1.06)	0.339
≤34 years	66	41	1 (Ref)	
>34 years	39	17	0.47 (0.21–1.05)	0.067
BMI	105	58	0.98 (0.88–1.10)	0.734
AMH	91	50	1.18 (0.98–1.44)	0.087
Duration of stimulation	105	58	0.9 (0.76–1.05)	0.172
E2 level at triggering day	104	57	0.99 (0.99–1.01)	0.229
Total FSH cumulative dose	105	58	0.99 (0.99–1.01)	0.615
Stage	100	55		
I	51	29	1 (Ref)	
II	32	15	0.67 (0.28–1.63)	0.376
III	17	11	1.39 (0.45–4.34)	0.570
Histotype	103	56		
Duttal	94	49	1 (Ref)	
Lobular and Other	9	7	3.21 (0.63–16.29)	0.158
Grade of differentiation	95	53		
1	5	4	1 (Ref)	
2	39	18	0.21 (0.02–2.09)	0.185
3	51	31	0.39 (0.04–3.72)	0.411
BRCA				
Wild-type versus mutated/VUS	95	50	2.11 (0.76–5.83)	0.149
BRCA1 versus wild-type/BRCA2	87	44	1.33 (0.28–6.34)	0.718
Hormone receptors	103	57		
ER+ and PR+	75	39	1 (Ref)	
ER + and PR -	10	7	2.15 (0.52–8.97)	0.292
ER- and PR -	18	11	1.45 (0.51–4.15)	0.488
HER-2 expression	100	56		
Absent	83	47	1 (Ref)	
Present	17	9	0.86 (0.3–2.45)	0.780
Triple negative	99	56		
No	87	49	1 (Ref)	
Yes	13	7	0.88 (0.27–2.84)	0.832

Bold font highlights statistically significant differences. CI: confidence interval. BMI: body mass index. AMH: anti-Mullerian hormone. E2: estradiol. FSH: follicle-stimulating hormone. BRCA: breast cancer gene. VUS: variants of uncertain significance. ER: estrogen receptor. PR: progesterone receptor. HER-2: human epidermal growth factor receptor 2.

## Data Availability

The data presented in this study are available in this article.

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
