# Peer review of "Oocyte Quality Assessment in Breast Cancer: Implications for Fertility Preservation"

_cancers, 2022, doi:10.3390/cancers14225718_

Round 1

Reviewer 1 Report

In this manuscript authors evaluated the effects of breast cancer on ovarian response and on oocytes quality in controlled ovarian hyperstimulation (COH). They found that breast cancer was not associated with an impairment of ovarian reserve, but it was an important risk factor of presence of dysmorphic oocytes.  

The manuscript is clear and generally well written. I have only two suggestions. In particular:

Introduction: it deserves to be specified that oocytes can also be damaged by the cryopreservation process itself. In fact, this process indice ROS production that damage the oocytes (see PMID: 35453348 ). This is an important point to add since an appropriate oocyte preservation protocol is essential to preserve fertility in young women undergoing chemotherapy

Figure 2: Scale bar should be added 

Reviewer 2 Report

The paper is interesting and in accord with the literature is quite complete.

Contribution of all authors is significant and could be interesting for scientist.

Analysis and data interpretation are adequacy, writing style could be improving.

The only suggestion is you could add a drawing this it would make reading the manuscript more appealing.

Moreover, I suggest improving how a tailed and multidisciplinary approach is needed by citing:

PMID: 34268981 

PMID: 34706754

Reviewer 3 Report

This is a well written study assessing the oocyte dysmorphism in breast cancer patients, compared to healthy patients.

There is a lot of literature supporting oocyte freezing for breast cancer patients and the above paper does not add much to the current literature

However, it is well designed and makes a good read

Author Response

In accordance with the reviewers’ suggestions, the revised manuscript was checked by Language Editing Services -  MDPI's Author Services for English proofreading.